# Gender Bias and the Lack of Equity in Pandemic Nursing in China: A Qualitative Study

**DOI:** 10.3390/ijerph181910273

**Published:** 2021-09-29

**Authors:** Pingting Zhu, Qiwei Wu, Xinyi Liu, Ericka Waidley, Qiaoying Ji, Ting Xu

**Affiliations:** 1School of Nursing, Yangzhou University, Yangzhou 225009, China; MZ120201760@yzu.edu.cn (Q.W.); MX120190879@yzu.edu.cn (X.L.); MZ120201761@yzu.edu.cn (Q.J.); MX120201102@yzu.edu.cn (T.X.); 2Jiangsu Key Laboratory of Zoonosis, 136 Jiangyang Middle Road, Yangzhou 225009, China; 3School of Nursing, Linfield University, 2900 NE 132nd Ave, Portland, OR 97230, USA

**Keywords:** COVID-19, sexism, female nurses, gender equity, gender bias, female stereotype, pandemic

## Abstract

There has long been a gender bias in medicine. This qualitative study aims to identify the experience of sexism among frontline female nurses and further explore their expectations and possible strategies to get rid of gender bias. This is a descriptive phenomenological study of 23 female nurses with 11 ± 3.98 years of experience who spent 36 ± 6.50 days at the frontline during the initial COVID-19 outbreak. We employed Colaizzi’s phenomenological analysis method to understand the subjective experiences, revealing the following themes: (a) materialization of gender identity; (b) incoordinate relationships; (c) future voice of female nurses. The gender bias experienced by female frontline nurses further challenges their emotional identity and self-identity. Therefore, it is important to require extensive consciousness-raising and policy support to defend female nurses’ rights.

## 1. Introduction

The eye of the COVID-19 storm caused deadly destruction in Wuhan that was visible throughout the world. Doctors and nurses from all over the country joined the efforts to fight against the epidemic [1]. Female nurses accounted for two-thirds of the total number of the medical support team [2]. These female nurses were efficient in delivering care to the COVID-19 patients and assisted in stabilizing the pandemic in Wuhan [3,4]. One year has passed since the beginning of the pandemic in Wuhan, and the recognition of the female nurses appears to have gradually disappeared. However, while watching the pandemic unfold, it increased the awareness of the issue of women’s interests post-disaster. An example of this awareness is the Beijing Declaration and Platform for Action in 2020 which highlighted that equality between women and men is a fundamental condition for social justice and sustainable development. It is regarded as a milestone all over the world for gender equality and the holistic development of women. However, even with these developments, which are regarded as benchmarks for change, women’s rights issues are being ignored [5].

Medicine is traditionally regarded as a male-dominated field, which may be a universal consensus worldwide [6]. This conception is not about gender ratios, as in fact, there are far more women than men in the medical field, but refers to a dominant cultural form based on a particular kind of logic that embraces heroism and rationalism [7]. Previous research pointed to “the Matilda effect” to describe a bias denying recognition within the medical field [8,9]. Even with the female-dominated response to the pandemic, the traditional gender gap issues in healthcare continue to exist [10,11]. During the epidemic, in numerous reports praising frontline female nurses, we would have thought that the performance of women would be the recognition needed to boost the culmination and long-term benefits of feminism. However, the peak that occurred during the pandemic was just that, a single occurrence. Then, it fell off a cliff, below the original level, and got stuck in stigmatization [12].

There are still many people who believe that compassion and caring for patients is driven by the cogs of femininity [13]. Such unconscious, nonmalicious biases and gender expectations also existed behind the widespread coverage of Wuhan. Most frontline real-time professional spokespersons tend to be men, which to a certain extent conceals women’s professional and leadership abilities and also suppresses women’s voices. Statistically, more than four out of five of the COVID-19 decision-makers were identified as men [14]. In contrast, when the photo lens cut to the picture of women, it tended to show facial blisters and bloodstains squeezed out by masks, rather than expressions of expertise and confidence in providing care. Therefore, in such an ultra-high-risk environment, men, who are a minority of nurses, appear to have a higher authority than women, which seems to be a declaration of frontline sovereignty and public conviction [15]. In this research, female nurses who had volunteered to go to Wuhan were interviewed. The aim of this study is to investigate the gender bias experienced by female nurses during and after COVID-19 supporting assignments.

## 2. Materials and Methods

### 2.1. Research Design

This research adopted interpretative phenomenological analysis to understand the subjective experiences and the reasons for these experiences amongst female nurses [16,17]. Critical qualitative content analysis was reported according to the COREQ checklist.

### 2.2. Participants and Recruitment

Participants who had work experience in the COVID-19 outbreak in Wuhan were recruited using a purposive snowball strategy through the Jiangsu Commission of Health. The inclusion criteria for participants consisted of the following: (1) was a volunteer participant when the epidemic broke out in Wuhan, (2) female, and (3) over the age of 18 years. The exclusion criteria were male nurses and an inability to complete the interview. To determine the number of the sample in qualitative research, and considering the determination of the number of the sample, data saturation (no more new information emerging) was found to be achievable with 16 female nurses (*n* = 23) [18].

### 2.3. Data Collection

The study data were collected from September to November 2020. The data collection used a semistructured interview methodology with participants from three hospitals in Yangzhou, Jiangsu province. We used it to guide interviews after a literature review of a large number of feminist theories. Nevertheless, there is no single feminist theory or singular understanding of feminism. However, there is certainly a starting point of feminist theory: “feminism is a movement to end sexism, sexist exploitation, and oppression” [6,19,20,21,22]. A semistructured interview is a common mode in phenomenological research that helps to remove gender bias in research by allowing women to express their ideas, opinions, or experiences in their own language, while capturing women’s voices, especially those in line with feminist ideals [6,23]. It consisted of open-ended questions (Table 1). Field testing for potential participants was conducted, and the questions were modified prior to the formal application of the interview questions [24]. Researchers strive to establish trust and a comfortable atmosphere before entering the topic and starting the interview with a regular conversation (e.g., “How is your day?”). Participants enter a relaxed state where rapport is developed over time. Each interview was conducted for approximately 60–90 min in the meeting room of the participants’ hospital.

### 2.4. Data Analysis

Transcribed interview data were imported into qualitative analysis software Nvivo Version 12. Data were analyzed using Colaizzi’s phenomenological analysis method [25,26]. The transcripts were read carefully several times by researchers, and then initial codes, categories, and subcategories were identified. Three main themes were formed by describing the categories in detail. Several strategies were used to ensure trustworthiness and credibility. The transcripts were sent to the participants to confirm the accuracy of the transcribed content. The coding and categorization processes were performed independently by the two researchers. The researchers had regular and continuing discussions to verify the appropriateness of the conceptual meanings and terminology. The audit trail was maintained to ensure all analysis steps could be traced back to original interviews. All research results were presented through the consensus reached by the whole team after discussion.

### 2.5. Ethical Considerations

The Ethics Committee of the School of Nursing of Yangzhou University approved the study (IR code: YZUHL2020002). Prior to data collection, the researchers provided a self-introduction to the participants and explained the study objectives and methodology. Verbal (on WeChat) and/or written (face to face) informed consent was obtained from all participants. All principles of confidentiality and anonymity were maintained during the study. The participants were told they had the freedom to withdraw from the study at any time without consequences.

## 3. Results

The study cohort included 23 female nurses who had worked as frontline nurses in Wuhan during the COVID-19 epidemic in 2019. The mean age of participants was 32 years old with a mean work experience of 11 years. All participants were registered nurses. The demographic characteristics of the participants are presented in Table 2.

Three themes were extracted from the content of the interviews: materialization of gender identity, incoordinate correlations, and future voice of female nurses.

### 3.1. Theme 1—Materialization of Gender Identity

Two-thirds of the interviewees expressed experiences of suffering materialization of gender identity. Various examples were given by the participants. We divided this theme into three subthemes: salutation should not be gendered, symbolization of “head shaving”, and substitution of identity.

#### 3.1.1. Salutation Should Not Be Gendered

Participants felt that media professionals often create attention-grabbing words for high exposure and that these words were often offensive because they were the product of the combination with gender, such as “female frontline soldier”. For participants with the same frontline experience, gendered salutations were infuriating.

“It’s really uncomfortable to be particularly emphasized on gender issues, and there is no need to emphasize gender in the same experience.” (Participant 15)

“Women do need to overcome more difficulties at the emotional and physical level compared with men in the special environment at the frontline of the epidemic. These combined word collocations seem to praise a woman, but the excessive emphasis on gender is a misunderstanding of these praise words.” (Participant 11)

#### 3.1.2. Symbolization of “Head Shaving”

Participants explained that head shaving was done in the beginning of the epidemic to reduce the chance of becoming infected when little was known about COVID-19. Because they were also afraid of unknown viruses, head shaving was easier to accept compared to the risk of infection. Some participants said that the use of “head shaving” to highlight their contributions to the frontline was a gender bias stemming from a male perspective in portraying female nurses.

“The media showed the photos of female nurses with shaved hair in public view. It was shocking at first, and it was a subversion of the image of women. But it’s been reposted over and over again and it’s not just visual fatigue, and I started wondering is that all we can do?” (Participant 8)

Others say that “shaving and supporting” was a sense of ritual in their heart and that the sense of the ritual was not to care about whether or not to shave their beautiful hair but that afterward they were ready to fight the unknown virus.

“Everyone else had their head shaving, so I did the same. It seemed that the mission to join the frontline has begun at that time.” (Participant 2)

#### 3.1.3. Substitution of Social Identity

As the epidemic in China unfolded, the focus on the female nurses tended to paint a picture of these professionals as wives and mothers versus one of experienced healthcare professionals willing to be deployed on a moment’s notice to care for COVID patients. Some participants felt it was “ignoring their expertise” because they clearly knew that some of their original social roles would not be fulfilled if they chose to go to the frontline.

“It is a personal choice to go to the frontline for support, and the media likes to use special headlines related to family roles to exaggerate the atmosphere.” (Participant 4)

One participant said that these headlines related to social roles were involuntarily touching, but the growing number of reports made her feel guilty about her family because media coverage of the role of a mother or wife was a stimulus, a source of empathy.

“There is no need to use females’ other social roles to exaggerate the situation. It would be a handicap.” (Participant 12)

### 3.2. Theme 2—Incoordinate Correlations

Female stereotypes often lead to preconceptions that lead to incoordinate correlations. We divided this theme into two subthemes: professionalism is not the same as gender characteristics, arbitrary definitions using female stereotypes.

#### 3.2.1. Professionalism Is Not the Same as Gender Characteristics

Some participants said that in caring for COVID-19 patients, being humane and talking softly was perceived by patients as feminine rather than as a nurse. Even if this understanding is encountered in the day-to-day work, in the frontline, they wanted to be seen as a nurse who was willing to take risks to care for them, rather than as a woman, because defining them by their gender inevitably accentuates their feminine characteristics.

“In the consensus of the public, women represent gentleness, but it does not mean that I will choose a career because of my gender.” (Participant 7)

Participants were exposed to the popular belief that men were equal to doctors and women to nurses. Because everyone in the shelter was wearing the same personal protective clothing, patients identified gender by height and voice. The gender was determined only to confirm whether the medical staff should be addressed as doctor or nurse. People’s subconscious perception of doctors and nurses was mentioned.

“Even in the subconscious of the nation, the difference between men and women among medical staff is mainly the difference between doctors and nurses, thinking that we are secondary.” (Participant 6)

“When it’s a man, patients call him doctor, but he’s a male nurse.” (Participant 21)

#### 3.2.2. Arbitrary Definitions Using Female Stereotypes

In describing the experience, one participant said that if a male nurse was administering an infusion to a patient and the patient felt pain, the patient would just say “forget it”. As for the female nurse, she would be accused of not being gentle enough.

“Why should we emphasize the distinction between men and women? This is a very strange phenomenon. After all, we do all the same things.” (Participant 1)

Family stereotypes of the female even influence their career choices.

“Nursing was the career my family chosen for me because they thought it was a suitable job for a girl. I think that’s probably because of ‘inherently feminine trait’.” (Participant 3)

### 3.3. Theme 3—Future Voice of Female Nurses

Reality often shows the circumstances of female nurses in a cruel way [27]. However, as more female nurses receive the recognition they deserve, their voices are becoming stronger, and they are beginning to voice their dissatisfaction and demands [28].

#### 3.3.1. The Need to Remove Gender-Colored Spectacles

Not being trusted and not feeling involved at work was described as an “invisible barrier” by participants, especially when they mentioned that the patient’s symptoms and discomfort could be related to psychological causes, which was perceived as a lack of expertise because it has been assumed that only females talk about psychological issues.

“Raising questions or comments at work were not being acknowledged. But I think compared with men, women have no shortage of knowledge.” (Participant 16)

#### 3.3.2. Career Development

Consideration for career advancement would not be a priority if women were not involved in specific tasks, such as frontline supporting.

“In the past, leaders would give male colleagues more opportunities for continuing education. Since returning from Wuhan, I seem to be the first in line to get some chances to learn, such as the training of specialist nurses.” (Participant 2)

Participants mentioned that a female’s fertility could be a career uncertainty and that pregnant nurses were treated like “ball players” by various departments because they were unable to have night shifts and were taken on a heavy workload.

“The head nurse will propose to the human resources department that male nurses are needed, after all, they do not need maternity leave.” (Participant 3)

Men still hold essential positions on temporary assignments (frontline working in Wuhan).

“The spokesmen and team leaders are all men.” (Participant 19)

#### 3.3.3. Creating a Gender-Friendly Working Environment

In the early phase of COVID-19, feminine products were deficient. It was difficult for female workers to have access to feminine napkins, tampons, and other comfort measures during deployment. Even though China had a “green channel” for emergency supplies, these feminine products were not available at first.

“I heard that those who have just arrived in Wuhan to support were not provided with enough female products because all material deliveries are based on protective equipment. But women make up the majority of nurses population, and women’s supplies should also be given priority.” (Participant 5)

As for supporting dangerous areas like Wuhan, the medical condition of female nurses should be checked before they go.

“One team member went to Wuhan and found out that she was pregnant, so we did not let her into the quarantine area. Other medical teams had similar situations and all of them were (flown)… back to Yangzhou.” (Participant 14)

## 4. Discussion

The purpose of this study was to investigate the gender bias and inequality experienced by female nurses during and after supporting the Wuhan assignments.

Gender inequality is a major challenge to global health, and the existence of gender inequalities and prejudices impedes the delivery of effective and efficient healthcare services by health systems [29,30]. The foundation of the health system includes achieving goals of universal health and sustainable development, and if gender equality is not achieved, then building on a shaky foundation is not sustainable. Embedding a series of gender biases in the health system can greatly undermine female’s enthusiasm and work ability and chips away at the foundation of patient care services [31]. Similarly, in a traditional hierarchical medical environment, there is concern that a culture of bullying is acceptable at the lower levels of the pyramid. In addition, female nurses who have just started work are subject not only to gender bias, but also to vertical workplace bullying [32,33,34]. If allowed to continue, these behaviors create a culture of distrust and fear of retribution and contribute to high turnover rates in many institutions. High turnover can shake up the healthcare industry, especially in the event of a pandemic. It was, therefore, important to know whether female nurses who render care in Wuhan experienced gender bias and their feelings, as this will have implications for future recruitment rates of female nurses in the medical profession and support rates after the outbreak.

Whether gender bias occurs at the frontline or after the supporting mission is over (e.g., deployment to the front lines of the pandemic), the contradictions in roles and recognition are unavoidable. From a macro level, it is closely related to cultural background, social process, and social ideology. Although the status of Chinese women is consistently being scrutinized throughout the world, there continues to be an intermittent influence of traditional values that affect the evolution of their role. The mainstream traditional belief in China is Confucianism, which evolved after the washing of history and the subsequent ideological revolution. Many Chinese are trying to abandon the ideological dregs of “men are superior to women” and “men farming and women weaving” [35]. However, Confucianism has an unshakable position in the long history of China, and the side effects of Confucianism are still detrimental to women [36].

Female nurses experienced a gender bias in vocabulary that could be found in this study. The media reports, which emphasized gender symbols, were seen as a sign of disrespect. Female nurses were portrayed as an expectant ideology using the materialization of gender identity. Previous studies have shown that high rates of materialization of women are commonly used in the media, for example, by highlighting women’s bodies in advertising [37]. The materialization of women in media has long been argued to affect men’s attitudes in ways that could lower the social value of women [37,38]. Clearly, media reports are not correct for frontline nurses, who are expected to demonstrate professional medical skills during their frontline experiences rather than rely on their appearance and identities to be considered heroes of the moment. Such stereotyped female power roles are the embodiment of the ideology of patriarchal society [39]. The media tends to commoditize this gendered power, but in fact, women need to be seen as individuals who can speak and be heard.

“Femininity” and “nurse” always seem to be matching words. Previous studies have shown that patients experience a lot of confusion when presented with male nurses [40,41]. Moreover, male nurses can gain more tolerance, which is a new difference in this study. A systematic review and narrative synthesis study reported that the public’s impression of nursing is uneven [42]. Some people believe that nurse specialists are more knowledgeable than general practitioners and that they are also medical staff who can be contacted first, while others believe that nurses are followers of doctors and have not received higher education [42]. Research in the United Kingdom shows that parents are less likely to encourage nursing as a career choice [43], and a survey of public perceptions of nursing conducted in the United States reported that nursing was highly respected as a career [44]. In this study, nursing was found to be an occupation with feminine characteristics recommended by parents to their daughters. Thus, gender bias has been linked with preferences supportive of traditional gender roles.

The so-called “gender equality paradox” is the fact that gender segregation occurs across occupations. In a field often thought to be dominated by men, the medical domain, women are often seen as playing only a supporting role. It will take 99.5 years to close the global gender gap, even though the majority of the world’s healthcare professionals are female [5]. Female nurses present a pyramidal position distribution that indicates that only a small number of women are at the decision-making level at the top of the pyramid, while the larger base group is at the bottom of the pyramid and accounts for most grassroots workers [13,45]. Due to this distribution, healthcare leaders may prioritize male nurses for professional development opportunities and job promotion. The deployment of female nurses during the pandemic may indicate that changes in gender equality in China remain unclear; this is because participants were unsure whether increased opportunities were due to awareness of gender equality or to being on the front lines.

The passive gender schema is integrated into the entire external environment that is unfavorable to women. Studies have shown that the reasons for women’s career barriers in the workplace are believed to be psychological factors such as dependency and obedience [32]. Therefore, women are also labeled as sensitive and fragile, not strong enough, lacking the spirit of hardship, and generally unable to withstand high pressure in management positions [46]. On the contrary, it is these labels that make people think that caring for patients is a bounden duty commensurate with the female character. Women’s career development is often referred to as the “glass ceiling” to indicate invisible obstacles [47]. Previous studies have shown that this phenomenon is caused by the linkage of many factors [48,49]. These factors can include stereotypes, maternal stress (e.g., child-rearing and undertaking housework), and lack of role models for high-ranking women. As the accumulation effect of disadvantages increases, the ceiling continues to be a realistic barrier that cannot be broken through. In order to make small cracks in this barrier, it is important to create a work environment suitable for women.

There were some limitations in this study. First, it was conducted only for healthcare professionals in Yangzhou city, limiting the representativeness of the sample and the generalizability of the results. Second, quantitative approaches, which can be employed to measure perceived gender bias in female nurses, were not components of this study.

## 5. Conclusions

Current research on gender bias has focused chiefly on surgery departments, with few studies focusing on the experience of female nurses during the pandemic. This study adds to new evidence of gender bias and inequities in medicine. Using this new evidence to speak out for female power and using tension and contradiction as a driving force will facilitate further thought about these issues. Continuing research on women’s roles and rights in society, specifically in healthcare environments, provides an inexhaustible driving force for narrowing the distance in gender equality.

## Figures and Tables

**Table 1 ijerph-18-10273-t001:** Interview questions.

	Open-Ended Questions
1.	Can you talk about your experience of supporting Wuhan?
2.	During your support period, have you ever been subjected to gender bias? Could you please specify.
3.	What does gender equality mean to you?
4.	How do you feel about your career?
5.	What barriers/difficulties related to your gender if any have you faced during your carrier advancement?
6.	How did you feel when you were confronted with the barriers?
7.	What do you think of the phenomenon of gender bias?

**Table 2 ijerph-18-10273-t002:** Demographic characteristics of the study participants.

	Mean ± SD or *n* (%)
Age (years)	32 ± 3.80
Clinical work experience (years)	11 ± 3.98
Length of time in Wuhan (days)	36 ± 6.50
Marital status	
Married	17 (73.9)
Single	6 (26.1)

## Data Availability

The data presented in this study are available on request from the corresponding author.

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
