# Peer review of "Gender Bias and the Lack of Equity in Pandemic Nursing in China: A Qualitative Study"

_ijerph, 2021, doi:10.3390/ijerph181910273_

Round 1

Reviewer 1 Report

Congratulations on a great paper!

Author Response

It is a great honor to receive your recognition of this research. Thank you very much!

Reviewer 2 Report

Thank you for revising the paper, please have another revision to the English style and editing particularly on the new added parts.

Author Response

As the reviewer suggested that the English language and style need to be improved, we have invited professional editing organizations to check and modify our manuscript, especially the new added parts. Detailed revisions are highlighted in red and can be seen in the manuscript.

Reviewer 3 Report

Review of “Gender bias and the lack of equality in pandemic nursing in China”

23 Sept 2021

This manuscript describes a qualitative interview study looking at the experience of female nurses who volunteered on the front lines in Wuhan.

Overall the study is interesting. I think that the results are useful in showing that gender bias in nursing is still very prevalent. I did not find anything strongly relating to the population of frontline nurses, as opposed to general gender bias for nurses. But it is still a useful article.

I have only a few comments to improve the paper.

Lines 91-92: Researchers established trust with the participants, but readers will want to know how this happened. A very short description of how this was conducted would be helpful.

Figure 1: I don’t think that this figure adds value to the manuscript, and I think it could be removed. If it is kept, then it needs further explanation. Use of colors, what “advanced” means, etc. But my preference would be to remove the figure.

Line 311: the authors added “which is the same in this study” but I don’t believe that this study shows that women are limited to lower level. If there is evidence for this statement, please elaborate. Or just remove the statement. I think that Participant 2’s comment in lines 216-218 actually provides evidence against your statement.

These are minor issues, and with further English editing and some clarifications, this article will be ready to be published.

Author Response

Thank you for reviewing the manuscript, and the reply was uploaded as an attachment.

This manuscript is a resubmission of an earlier submission. The following is a list of the peer review reports and author responses from that submission.

Round 1

Reviewer 1 Report

Please see attached word document for comments 

Reviewer 2 Report

Overall, the article discusses an important and relevant topic, with appropriate research design and methodology, easy to follow argumentation, and the results offered are novel and inspiring. English language and style are relatively good, however some, probably unintentional, omissions, sometimes create problems in understanding certain sentences.

Minor revisions refered only to English editing and style as I suggested in my comment. Just one quick hand by a professional language editor and/or native speaker will do.

Reviewer 3 Report

The paper addresses gender inequality from an interesting and innovative viewpoint.

In order to make the readers to fully understand the points analysed in the paper, the authors should, however pay more attention to the following points: 

Results

The process leading to the identification of the codes and the themes could be expanded; this would allow the better visualise the authors' analytical thought and would at the same time give an explanation of fig 1 which in the form presented is not very clear. It would also more congenial to move lines 117-120 to above fig 1.

Themes

For all 3 themes (and sub themes) it would help if additional references to the participants' narrative is provided. Particularly for sub-themes 3.1.1 and 3.1.2 whose narrative do not seem clearly justify the theme identified. Probably explaining how the themes have emerged from the identified codes could improve this section.

Discussion:

This section seems lacking continuity with the previous part, very little reference is made to the themes identified a little justification is given with reference to the participants' narrative. For example, lines 290-292 would gain if an extract from the participants’ interviews is added.

Editing:

Moderate English revision is recommended e. g. lines 50; 78-80; 89; 92 just to suggest a few